# An Experimental Study of the Relation between Mode I Fracture Toughness, *K_Ic_*, and Critical Energy Release Rate, *G_Ic_*

**DOI:** 10.3390/ma16031056

**Published:** 2023-01-25

**Authors:** Yang Qiao, Zong-Xian Zhang, Sheng Zhang

**Affiliations:** 1Oulu Mining School, University of Oulu, 90570 Oulu, Finland; 2School of Energy Science and Engineering, Henan Polytechnic University, Jiaozuo 454099, China

**Keywords:** critical energy release rate, mode I fracture toughness, relation

## Abstract

The construction of the relation between the critical energy release rate, $G_{Ic}$, and the mode I fracture toughness, $K_{Ic}$, is of great significance for understanding the fracture mechanism and facilitating its application in engineering. In this study, fracture experiments using NSCB and CCCD specimens were conducted. The effects of specimen sizes, loading rate and lithology on the relation between $G_{Ic}$ and $K_{Ic}$ were studied. $G_{Ic}\;$was calculated by integrating the load–displacement curve according to Irwin’s approach. Based on the measured $K_{Ic}$ and $G_{Ic}$ of the rock specimens, a relation between $G_{Ic}\;$and $K_{Ic}$ was found to be different from the classical formula under linear elasticity. It was found that both specimen size and loading rate do not influence this relation.

## 1. Introduction

Fracture toughness and critical energy release rate are very important parameters in fracture mechanics. The construction of the relation between $G_{Ic}$ and $K_{Ic}$ is of great significance for understanding the fracture mechanism and establishing the relation between some fracture parameters such as the J-Integral [1], R-value [2] and crack propagation velocity [3,4]. It also facilitates the application of both values in numerical simulations and engineering. In fracture mechanics, the stress intensity factor characterizes the stress and displacement distribution of a pre-crack tip, while the energy release rate, *G*, refers to the rate of potential energy within the crack area [5]. Their relation is [6]:(1)$$G=K_{I}^{2}/E\;\mathrm{for}\;\mathrm{plane}\;\mathrm{stress}$$
or
(2)$$G=(1-v^{2})K_{I}^{2}/E\;\mathrm{for}\;\mathrm{plane}\;\mathrm{strain}$$

Either equation is based on the stress and displacement solutions around a pre-crack tip [6,7], where *K* is the stress intensity factor of a crack tip. Here, *G* quantifies the net change in potential energy that accompanies an increment of crack extension; and *K* characterizes the stresses, strains and displacements near the crack tip. The energy release rate describes global behavior, while *K* is a local parameter [2]. As described in [2], if a material fails locally at some combination of stress and strain, then crack extension must occur at a critical *K* value. This critical value is called fracture toughness (such as $K_{Ic}$ in mode I fracture). Griffith (1921), according to the first law of thermodynamics, developed an energy balance theory, stating that a sufficient condition for crack extension was that the energy absorbed by the material was greater than the energy required to form the new fracture surface [5,8]. Since energy release rate is uniquely related to stress intensity, *G* also provides a single-parameter description of crack-tip conditions, and *G_c_* (such as $G_{Ic}$ in mode I fracture) is an alternative measure of toughness, which is also called critical energy release rate. In linear elastic conditions, $G_{Ic}$ and $K_{Ic}$ are correlated with each other by Equations (3) and (4) [2]:(3)$$G_{Ic}=K_{Ic}^{2}/E\;\mathrm{for}\;\mathrm{plane}\;\mathrm{stress}$$
or
(4)$$G_{Ic}=(1-v^{2})K_{Ic}^{2}/E\;\mathrm{for}\;\mathrm{plane}\;\mathrm{strain}$$

However, many materials such as rock, concrete and ceramics are not linear elastic [9,10,11] and a large number of experimental data have shown that $G_{Ic}$ and $K_{Ic}$ do not conform to the relation in Equation (2) [9,10,12,13,14]. For example, it was found that as the length of a crack increases, $G_{Ic}$ and $K_{Ic}^{2}/E$ show a different relation from Equation (2), e.g., when a/w is 0–0.1 (a is the length of the crack and w is the width of the specimen), $G_{Ic}$ and $K_{Ic}^{2}/E$ show a power function relation, and when a/w is 0.1–0.85, they are linearly related [15]. The energy release rate first increases and then tends to stabilize with the crack propagation [16].

Some numerical codes use the relation between $G_{Ic}$ and $K_{Ic}$ to judge the extension behavior of a crack. For example, based on the implementation of the displacement extrapolation method (DEM) and the strain energy density theory in a finite element code, the kinking angle is evaluated as a function of stress intensity factors at each crack increment length, and the mechanical behavior of inclined cracks is analyzed by evaluating the stress intensity factors [17]. In addition, the global energy-based method is proposed to determine the crack propagation length and the crack propagation direction, and this method is formulated within an X-FEM-based analysis model, leading to a variational formulation in terms of displacements, crack lengths and crack angles [18].

Both $K_{Ic}$ and $G_{Ic}$ have a wide range of applications. Zhang [19] proposed a new method, based on the energy release rate, to assess fracture toughness, *K_c_*. Compared to previous methods, the new method was more consistent with actual damage mechanism and it did not depend on a specific critical damage value. Bearman et al. have shown that a strong correlation exists between the fracture toughness, *K_c_*, and the energy consumption of a laboratory crusher used to crush rock, indicating that the relation between *K_c_* and *G_c_* may have practical application in the evaluation of crushing equipment [20].

Based on the above description, the aim of this study is to establish a relation between fracture toughness, $K_{Ic}$, and critical energy release rate, $G_{Ic}$, by using the measurement results from a total of 128 limestone and sandstone specimens. In contrast to classical theory and formulae of the relation, the coefficients of the new formula are adjusted by considering the true crack area as well as the ductile fracture properties of the rock material. In addition, the study presents the effects of specimen sizes, loading rate and lithology on the relation between $G_{Ic}$ and $K_{Ic}$. This leaves the formula open to a wider range of applications.

## 2. Materials and Methods

Sandstone and limestone were selected to conduct the fracture tests on and they were taken from two quarries in Sichuan and Henan provinces, China. The rock specimens include two configurations: notched semi-circular bending (NSCB) and center-cracked circular disk (CCCD), as shown Figure 1a. Their physical and mechanical parameters are shown in Table 1. The pre-cracks were cut by a diamond wire with a diameter of 0.2 mm, resulting in a width of around 0.3 mm of the pre-cracks. The size, number, loading rate and configuration of the specimens are shown in Table 2. The RTX-3000 rock mechanic testing machine produced by GCTS was used in the experiment, as shown in Figure 1b. In order to maintain accuracy, a 25 kN force sensor was equipped. The three-point bending loading method was used to load the NSCB specimen and the CCCD was directly compressed. The displacement control method was frame displacement control, with an accuracy of 0.25 mm and a resolution of 0.025 mm.

## 3. Results and Analysis

### 3.1. Fracture Toughness ($K_{Ic}$) and Critical Energy-Release Rate ($G_{Ic}$)

In rock materials and under static loading conditions, the stress intensity factor at the peak load of the specimen is recognized as the fracture toughness, $K_{Ic}$, and the energy release rate as the critical energy release rate, $G_{Ic}$. According to the ISRM suggested method [23], the fracture toughness, $K_{Ic}$, of NSCB specimens was calculated by Equation (5), and the result is given in Table 3. The fracture toughness of CCCD specimens was calculated by Equation (6) [24].
(5)$$K_{Ic}=\frac{P_{max}\sqrt{\pi a}}{2Rt}Y_{NSCB}$$
$$Y_{NSCB}=-1.297+9.516\cdot \left[S/\left(2R\right)\right]-\left\{0.47+16.457\cdot \left[S/\left(2R\right)\right]\right\}\alpha +\left\{1.07+34.401\left[S/\left(2R\right)\right]\right\}\alpha^{2}$$
α=a/R   S/(2R)=0.6, α=0.2
(6)$$K_{Ic}=\frac{P_{max}}{t\sqrt{2R}}Y_{CCCD}$$
$$Y_{CCCD}=\sqrt{\frac{2}{\pi }}\sqrt{\frac{\alpha }{1-\alpha }}\left(1-0.4964\alpha +1.5582\alpha^{2}-3.1818\alpha^{3}+10.0962\alpha^{4}-20.7782\alpha^{5}+20.1342\alpha^{6}-7.5067\alpha^{7}\right)$$

$Y_{CCCD}\;$can be calculated by numerical simulations and the $Y_{CCCD}$ in this study is from [22,24].

The critical energy release rate ($G_{Ic}$) was the energy dissipated in forming per unit crack surface area. The $G_{Ic}$ can be determined by three methods: the stress-intensity-factor method, the J-integral method and Petersson’s method (modified) [25]. In this study, the $G_{Ic}\;$was calculated by integrating the load–displacement curve (Figure 2) according to Irwin’s approach [26,27], as shown by Equation (7).
(7)$$G_{Ic}=\frac{W-U}{A_{0}}=\frac{1}{2R\left(1-\alpha \right)B}\sum_{i=1}^{n}(P_{i+1}+P_{i})\left(u_{i+1}-u_{i}\right)$$
where W is the total work done by the load and *U* is the strain energy stored in the specimen. $A_{0}$ is the nominal crack area, *n* is the total number of data points and *i* denotes the *i*-th data point. *P_i_* and *u_i_* are the corresponding load and displacement of the *i*-th data point, respectively.

As can be seen in Table 4, there is a significant size effect for both fracture toughness and critical energy release rate. Both decrease with decreasing size. The critical energy release rate of CCCD is significantly greater than that of NSCB. The fracture toughness of the sandstone specimens is significantly less discrete than that of the limestone with a range of 0.5–1.52 (standard deviation is 0.30), but the range of fracture toughness for sandstone is 0.36–0.69 (standard deviation is 0.07).

### 3.2. Relation between $G_{Ic}$ and $K_{Ic}$

Based on the experimental data of $G_{Ic}$ and $K_{Ic}^{2}/E$ in Table 5, we can obtain a regression equation between $G_{Ic}$ and $K_{Ic}$, as shown by Equation (8) and in Figure 3.
(8)$$G_{Ic}=3.09\;\frac{K_{Ic}^{2}}{E}$$

Obviously, Equation (8) is different from Equation (2) which is valid for linear elastic fracture, i.e., the coefficient in Equation (8) is 3.09, while that in Equation (2) is equal to 1. The difference between Equations (2) and (5) may be caused by two main reasons: (1) The rock in this study shows viscous and even ductile fracture behavior. As shown in Figure 4, the load–displacement curve exhibits a slow slope in the initial stages, peak and end of the loading, indicating that some of the energy absorbed by the specimen is used for plastic deformation and microcrack development in addition to crack extension [28,29]. However, the energies used the plastic deformation and the microcrack development are not excluded when calculating the critical energy release rate. (2) The nominal crack area, rather than true crack area, was used to determine $G_{Ic}$ in this study. Since the true surface area of a crack is much larger than the nominal area, Zhang and Ouchterlony have pointed out that the $G_{Ic}$ should be based on the true crack area [30].

### 3.3. Effects of Specimen Sizes, Loading Rate and Lithology on the Relation between $G_{Ic}$ and $K_{Ic}$

Five specimen sizes and five loading rates were involved in the experiments of this study and the experimental results of the relation between $G_{Ic}\;$and $K_{Ic}$ are presented in Figure 5a–e, showing that both specimen size and loading rate do not influence this relation. In other words, this relation is valid for all specimen sizes and loading rates involved in this study.

To investigate how lithology influences the relation between $G_{Ic}$ and $K_{Ic}$, the sandstone data in Table 4 are summarized in Figure 6. Clearly, sandstone is suitable for the relation between $G_{Ic}$ and $K_{Ic}$, meaning that this relation is valid for both sandstone and limestone used in this study.

## 4. Discussion

In this study, a relation between fracture toughness and critical energy release rate is obtained by analyzing data from 128 specimens. The effect of specimen size and loading rate on this regression equation is also explored. It provides the basis for further development of fracture theory suitable for quasi-brittle materials such as rocks. The relation between fracture toughness and critical energy release rate is conducive to refining fracture theories applicable to quasi-brittle materials such as rock. For example, it is well known that J = *G* = $K_{Ic}^{2}/E$ in linear elastic models. When the unloading that occurs during crack growth does not follow the same path as loading in a realistic situation, this equation does not hold true. However, this study may make it possible to calculate *G* for a nonlinear elastic condition in terms of the changes in the load–displacement curve with respect to crack length. Thus, this method may be applied directly to the computation of J in nonlinear elastic conditions. The R-value is also known as the resistance to crack extension and is constant for a material. The R-value, as measured by the ASTM standard, is limited by the size of the specimen and the R-value will vary with crack growth [16,31]. However, according to Griffith’s theory, the resistance to crack extension is the surface energy of the material. When the crack is steadily extending, *G* is equal to R. Based on the relation between energy release rate and fracture toughness derived in this paper, it is easy to calculate the resistance to crack extension, R.

In Table 3 and Table 4, there is a significant size effect on fracture toughness due to fracture process zone [32,33]. Additionally, the energy release rate is calculated using the nominal crack area rather than the true crack area. The cracked surfaces of rocks are very rough, so the true crack surface area is larger than the nominal crack area [30]. Therefore, if the fracture process zone is considered and the true crack area is measured, the result would perhaps be more accurate.

## 5. Conclusions

Fracture toughness and critical energy release rates are experimentally determined for 128 specimens. Based on the determined data of $G_{Ic}$ and $K_{Ic}$, a relation between these two fracture parameters is obtained, which is $G_{Ic}=3.09\;K_{Ic}^{2}/E$, with an R^2^ value of 0.97. This coefficient, 3.09, is greater than 1, the coefficient in the linear elastic fracture relation;This regression equation coefficient, 3.09, is greater than 1, the coefficient of the linear elastic fracture relation. The two of the reasons for this discrepancy are: (1) the $G_{Ic}$ is determined using the nominal crack area rather than the true crack area in this study and (2) the rock fracture is of non-linear-elastic rather than brittle fracture in this study;The effect of rock specimen size on the relation between $G_{Ic}$ and $K_{Ic}$ under static conditions is very small and it can be ignored. Similarly, the effect of loading rate on the relation between $G_{Ic}$ and $K_{Ic}$ under quasi-static conditions is also neglectable. The lithology does not seem to affect the relation between $G_{Ic}$ and $K_{Ic}$ under static conditions, but the result is based on only two types of rock.

## Figures and Tables

**Figure 1 materials-16-01056-f001:**
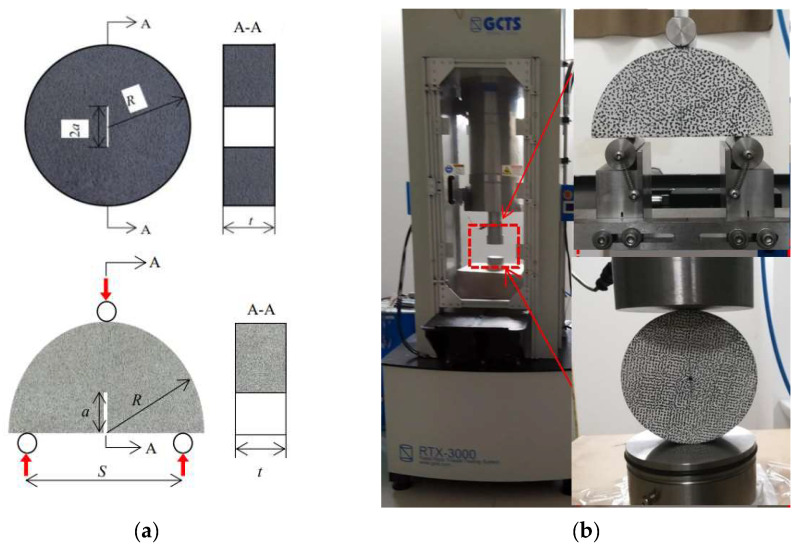
Configuration of rock specimens and testing machine. (**a**) Top: notched semi-circular bending (NSCB) specimens; (**b**) bottom: center-cracked circular disk (CCCD) specimens.

**Figure 2 materials-16-01056-f002:**
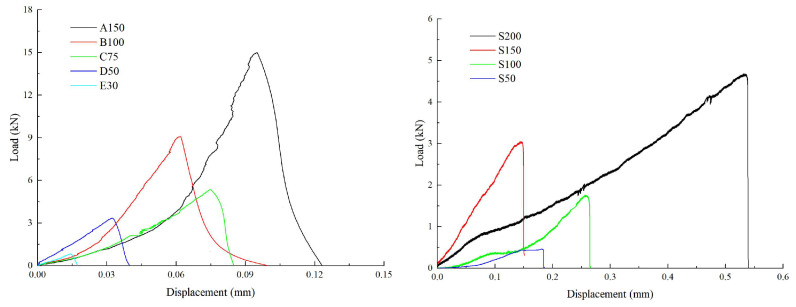
Load–displacement curve (the letters represent the group and the following number is the diameter of the specimen).

**Figure 3 materials-16-01056-f003:**
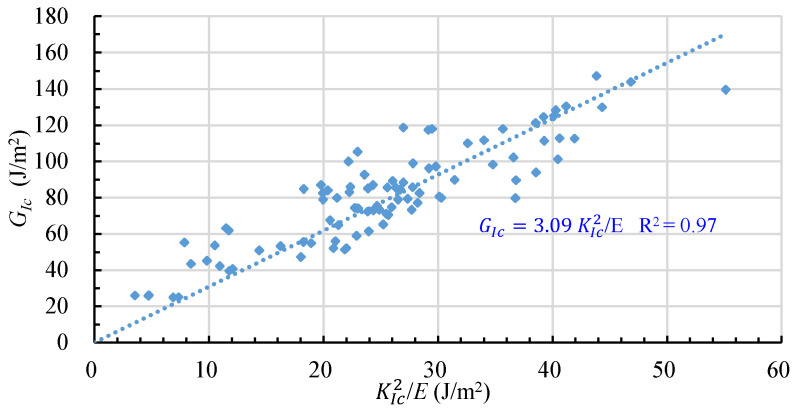
Relation between $G_{Ic}$ and $K_{Ic}^{2}/E$.

**Figure 4 materials-16-01056-f004:**
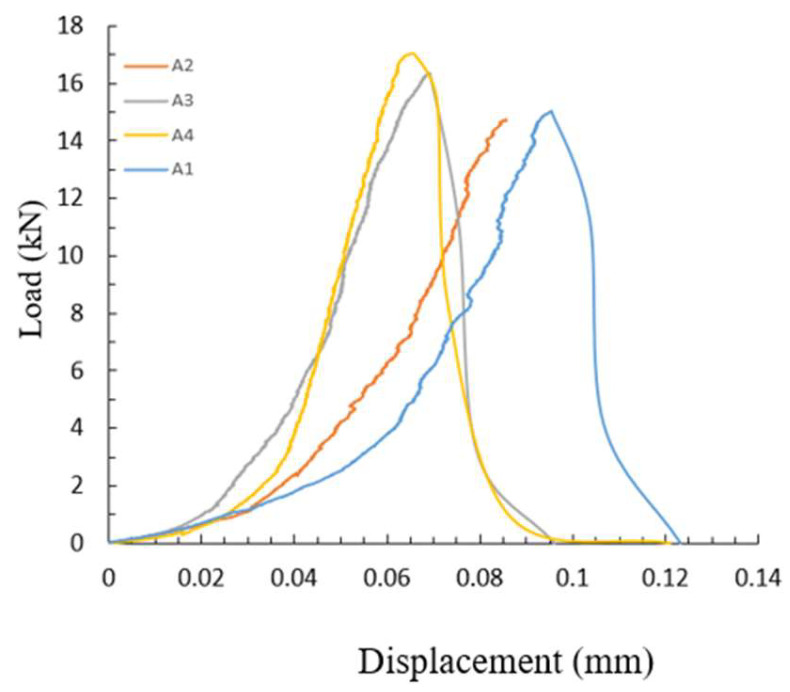
Load–displacement curve of limestone.

**Figure 5 materials-16-01056-f005:**
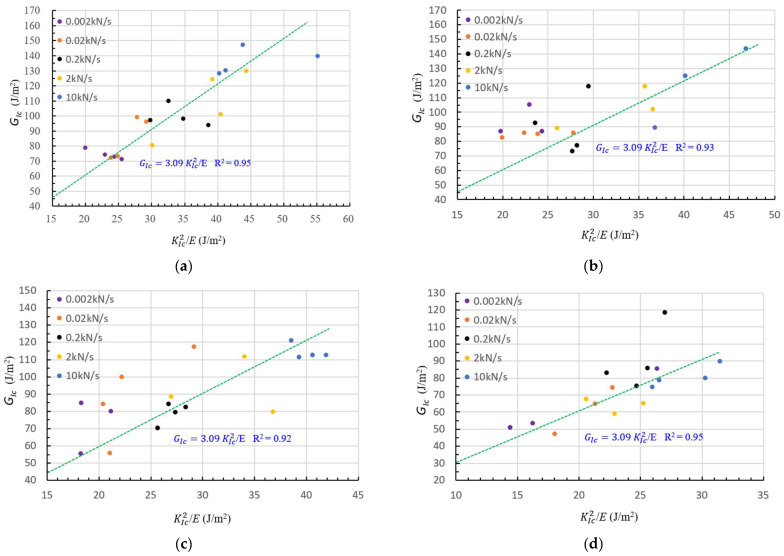
(**a**) Regression equation and data for *Φ* = 150 mm. (**b**) Regression equation and data for *Φ* = 100 mm. (**c**) Regression equation and data for *Φ =* 75 mm. (**d**) Regression equation and data for *Φ* = 50 mm. (**e**) Regression equation and data for *Φ* = 30 mm.

**Figure 6 materials-16-01056-f006:**
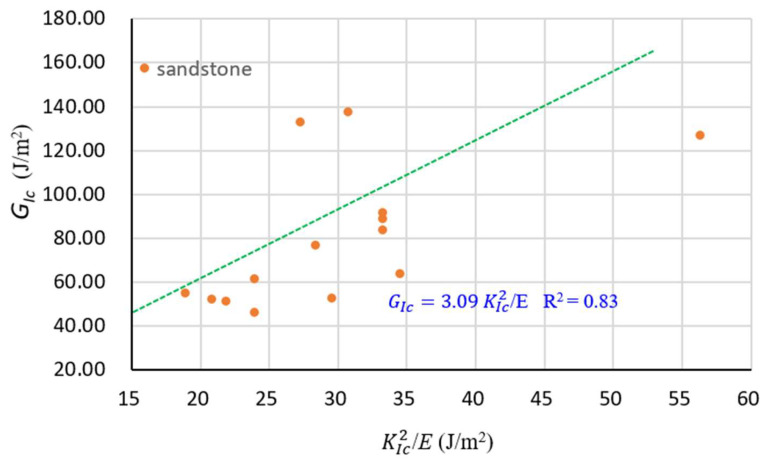
Regression equation and data of sandstone.

**Table 1 materials-16-01056-t001:** Physical and mechanical parameters of the tested rocks [21,22].

Parameter	Limestone	Sandstone
Uniaxial compressive strength (MPa)	178.80	58.53
Young’s modulus (GPa)	70.40	8.45
Poisson’s ratio	0.250	0.245

**Table 2 materials-16-01056-t002:** Sizes, number, loading rate and configuration of all specimens.

Rock	*Φ*/mm	*t*/mm	Total	Configuration	Specimen No.	Loading Rate
Limestone	150	60	20	NSCB	A1~A20	0.002~10 kN/s
Limestone	100	40	20	NSCB	B1~B20	0.002~10 kN/s
Limestone	75	30	20	NSCB	C1~C20	0.002~10 kN/s
Limestone	50	20	20	NSCB	D1~D20	0.002~10 kN/s
Limestone	30	12	20	NSCB	E1~E20	0.002~10 kN/s
Sandstone	200	60	4	NSCB	S200-1~S200-4	1.2 mm/s
Sandstone	150	45	4	NSCB	S150-1~S150-4	1.2 mm/s
Sandstone	100	30	4	NSCB	S100-1~S100-4	1.2 mm/s
Sandstone	50	15	4	NSCB	S50-1~S50-4	1.2 mm/s
Sandstone	200	60	3	CCCD	C200-1~C200-3	1.2 mm/s
Sandstone	150	45	3	CCCD	C150-1~C150-3	1.2 mm/s
Sandstone	100	30	3	CCCD	C100-1~C100-3	1.2 mm/s
Sandstone	50	15	3	CCCD	C50-1~C50-3	1.2 mm/s

Note: the ratio of crack length (a) to diameter is 0.4 for NSCB and 0.3 for CCCD.

**Table 3 materials-16-01056-t003:** Peak load, *P_max_*, and fracture toughness, $K_{Ic}$, of all specimens.

No.	*P_max_* (kN)	*K_Ic_* (MPa·m^1/2^)	No.	*P_max_* (kN)	*K_Ic_* (MPa·m^1/2^)	No.	*P_max_* (kN)	*K_Ic_* (MPa·m^1/2^)	No.	*P_max_* (kN)	*K_Ic_* (MPa·m^1/2^)
A1	15.016	1.186	B13	10.888	1.585	D5	2.968	1.225	E17	1.074	0.922
A2	16.318	1.271	B14	9.196	1.354	D6	3.034	1.265	E18	1.588	1.274
A3	16.340	1.310	B15	11.602	1.668	D7	2.782	1.127	E19	1.344	1.045
A4	17.014	1.340	B16	10.924	1.605	D8	2.810	1.161	E20	1.462	1.245
A5	17.912	1.400	B17	12.546	1.816	D9	3.382	1.378	S50-1	0.46	0.43
A6	18.258	1.434	B18	11.502	1.681	D10	3.338	1.342	S50-2	0.48	0.45
A7	16.220	1.296	B19	10.824	1.610	D11	3.060	1.252	S50-3	0.53	0.42
A8	16.826	1.324	B20	11.032	1.626	D12	3.222	1.318	S50-4	0.49	0.40
A9	18.520	1.449	C1	5.354	1.183	D13	3.268	1.333	S100-1	1.47	0.45
A10	19.288	1.515	C2	5.172	1.135	D14	2.900	1.204	S100-2	1.75	0.53
A11	19.866	1.565	C3	5.142	1.134	D15	3.144	1.27	S100-3	1.72	0.53
A12	21.046	1.648	C4	5.550	1.221	D16	3.868	1.584	S100-4	1.71	0.53
A13	22.188	1.767	C5	5.708	1.250	D17	3.588	1.488	S150-1	3.08	0.50
A14	20.930	1.662	C6	6.478	1.433	D18	3.586	1.460	S150-2	3.18	0.54
A15	18.506	1.456	C7	5.464	1.198	D19	3.240	1.366	S150-3	3.60	0.53
A16	21.164	1.688	C8	5.548	1.217	D20	3.240	1.352	S150-4	3.00	0.49
A17	21.422	1.684	C9	6.262	1.371	E1	0.572	0.501	S200-1	4.99	0.48
A18	22.392	1.757	C10	6.302	1.388	E2	0.684	0.564	S200-2	5.39	0.51
A19	21.504	1.703	C11	6.418	1.414	E3	0.826	0.720	S200-3	7.06	0.69
A20	25.032	1.970	C12	6.150	1.344	E4	0.670	0.578	S200-4	6.52	0.62
B1	9.104	1.309	C13	7.312	1.609	E5	0.696	0.577	C50-1	1.94	0.41
B2	8.170	1.180	C14	6.976	1.548	E6	0.894	0.744	C50-2	1.96	0.29
B3	8.798	1.272	C15	6.244	1.378	E7	0.990	0.832	C50-3	1.99	0.31
B4	7.534	1.121	C16	7.542	1.662	E8	0.828	0.726	C100-1	7.40	0.39
B5	9.002	1.297	C17	7.764	1.691	E9	1.008	0.861	C100-2	6.93	0.36
B6	8.202	1.185	C18	7.528	1.663	E10	0.942	0.771	C100-3	-	-
B7	9.332	1.255	C19	7.476	1.647	E11	0.788	0.696	C150-1	14.21	0.41
B8	9.512	1.399	C20	7.736	1.718	E12	1.092	0.878	C150-2	15.08	0.43
B9	9.992	1.441	D1	2.600	1.070	E13	1.088	0.910	C150-3	15.79	0.45
B10	8.990	1.289	D2	3.362	1.362	E14	1.112	0.900	C200-1	24.62	0.46
B11	9.218	1.397	D3	2.572	1.050	E15	1.092	0.911	C200-2	28.07	0.52
B12	9.396	1.410	D4	2.442	1.007	E16	1.196	1.039	C200-3	28.46	0.53

**Table 4 materials-16-01056-t004:** Statistical analysis of $G_{Ic}$ and $K_{Ic}$.

	A Group	B Group	C Group	D Group	E Group	S Group	CCCD
$\;G_{Ic}$ mean (J/m^2^)	100.86	96.84	90.61	73.68	43.62	82.51	296.42
$\;G_{Ic}\;$max (J/m^2^)	147.14	143.74	121.12	118.51	73.95	137.30	433.85
$\;G_{Ic}\;$min (J/m^2^)	71.10	73.31	55.54	47.22	25.00	51.27	193.43
$G_{Ic}\;$median (J/m^2^)	97.60	87.18	84.47	74.67	42.77	76.85	280.13
$K_{Ic}$ mean (MPa·m^1/2^)	1.52	1.42	1.41	1.28	0.83	0.5	0.44
$K_{Ic}\;$max (MPa·m^1/2^)	1.97	1.82	1.72	1.49	1.25	0.69	0.53
$K_{Ic}\;$min (MPa·m^1/2^)	1.19	1.19	1.13	1.01	0.50	0.40	0.36
$K_{Ic}\;$median (MPa·m^1/2^)	1.49	1.40	1.38	1.32	0.85	0.5	0.43

Note: A–D group (limestone) is NSCB with the diameters *Φ =* 150, 100, 75, 50 and 30 mm. S group (sandstone) is NSCB with the diameters *Φ =* 200, 150, 100 and 50 mm. CCCD specimen is sandstone with the diameters *Φ =* 200, 150, 100 and 50 mm.

**Table 5 materials-16-01056-t005:** $G_{Ic}$ and $K_{Ic}^{2}/E$ of all specimens.

No.	$K_{Ic}^{2}/E$ (J/m^2^)	$G_{Ic}$ (J/m^2^)	No.	$K_{Ic}^{2}/E$ (J/m^2^)	$G_{Ic}$ (J/m^2^)	No.	$K_{Ic}^{2}/E$ (J/m^2^)	$G_{Ic}$ (J/m^2^)	No.	$K_{Ic}^{2}/E$ (J/m^2^)	$G_{Ic}$ (J/m^2^)
A1	19.98	78.76	B13	35.69	117.91	D5	21.32	64.74	E17	12.08	40.55
A2	22.95	74.00	B14	26.05	89.18	D6	22.73	74.36	E18	23.06	73.95
A3	24.38	72.86	B15	39.53	161.64	D7	18.04	47.22	E19	-	-
A4	25.51	71.10	B16	36.60	102.05	D8	0.00	0.00	E20	22.02	52.02
A5	27.84	98.91	B17	46.85	143.74	D9	26.98	118.51	S50-1	21.88	51.27
A6	29.21	96.14	B18	40.14	124.95	D10	25.59	85.63	S50-2	23.96	46.18
A7	23.86	72.18	B19	36.82	89.50	D11	22.27	82.97	S50-3	20.88	52.04
A8	24.90	73.29	B20	37.56	68.36	D12	24.68	75.29	S50-4	18.93	54.78
A9	29.83	97.11	C1	19.88	104.79	D13	25.24	65.10	S100-1	23.96	61.36
A10	32.61	109.97	C2	18.30	84.70	D14	20.59	67.42	S100-2	-	-
A11	34.80	98.09	C3	18.27	55.54	D15	22.91	58.87	S100-3	33.24	91.41
A12	38.58	93.92	C4	21.18	79.89	D16	35.65	49.40	S100-4	33.24	88.71
A13	44.36	129.82	C5	22.20	99.92	D17	31.46	89.73	S150-1	29.59	52.43
A14	39.24	124.42	C6	29.17	117.39	D18	30.28	79.81	S150-2	34.51	63.50
A15	30.12	80.46	C7	20.39	84.07	D19	26.51	78.75	S150-3	33.24	83.57
A16	40.48	101.10	C8	21.04	55.88	D20	25.97	74.67	S150-4	28.41	76.85
A17	40.29	128.21	C9	26.70	84.25	E1	3.57	25.88	S200-1	27.27	132.60
A18	43.86	147.14	C10	27.37	79.27	E2	-	-	S200-2	30.78	137.30
A19	41.20	130.28	C11	28.40	82.43	E3	7.36	25.00	S200-3	56.34	126.85
A20	55.13	139.53	C12	25.66	70.40	E4	4.75	26.11	S200-4	-	-
B1	24.34	86.89	C13	36.78	79.61	E5	4.73	25.72	C50-1	19.89	193.43
B2	19.78	86.93	C14	34.04	111.65	E6	7.86	55.17	C50-2	-	-
B3	22.99	105.17	C15	26.98	88.33	E7	9.83	45.05	C50-3	-	-
B4	-	-	C16	39.24	76.89	E8	-	-	C100-1	18.00	200.93
B5	23.90	85.04	C17	40.62	112.67	E9	10.53	53.51	C100-2	15.34	242.54
B6	19.95	82.48	C18	39.29	111.28	E10	8.44	43.43	C100-3	-	-
B7	22.38	85.88	C19	38.54	121.12	E11	6.88	24.89	C150-1	19.89	280.13
B8	27.81	85.64	C20	41.93	112.51	E12	10.95	42.11	C150-2	21.88	269.24
B9	29.50	117.89	D1	16.27	53.27	E13	11.76	61.92	C150-3	23.96	305.30
B10	23.60	92.65	D2	26.35	85.38	E14	11.51	63.10	C200-1	25.04	396.72
B11	27.73	73.31	D3	-	-	E15	11.79	39.50	C200-2	32.00	433.85
B12	28.24	77.08	D4	14.41	50.84	E16	-	-	C200-3	33.24	345.61

## Data Availability

The data used to support the findings of this study are available from the corresponding author upon request.

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
