# Peer review of "An Experimental Study of the Relation between Mode I Fracture Toughness, KIc, and Critical Energy Release Rate, GIc"

_materials, 2023, doi:10.3390/ma16031056_

Round 1

Reviewer 1 Report

The manuscript has potential, but the authors are unable to show it.

The paper must contain nomenclature - all symbols, abbreviations and markings should be explained using nomenclature.

Please pay attention to the records of mathematical formulas and their numbering - sometimes it is done in an unclear way.

Please check very carefully the record of quantities where superscripts and subscripts are used. Already at the abstract level, the authors' oversight in this respect is visible - see line 13.

In the paper, please include technical drawings of both specimens used in experimental research, along with dimensions - these should be parameterized drawings, due to diameter and thickness.

The authors do not mention anything about the length of the crack - the fissure. Please provide its physical and relative dimensions, related to the overall dimensions of the specimens.

Please show in the pictures in more detail the arrangement of the specimens on the testing machine, let the readers see how the measuring apparatus, measuring sensors are attached - it must be visible - for both types of specimens.

Table 1 raises doubts. The physical units are "MPa" and "GPa", and not "Mpa" and "Gpa" as the authors state. Please absolutely correct this. Please provide information whether the authors themselves determined the values given in Table 1. If so, please provide the dimensions of the specimens, their technical drawings, information on the number of specimens used, statistical analysis of the results along with the dispersion. If they are literature data - please provide a reference to a specific scientific paper.

Please specify what the authors recorded during the test - force, displacement of the traverse or displacement of the point of application of the force or opening of the crack surface. The choice of size will affect the obtained results, the graphs that are missing in the authors' work and the final result. Please provide for each of the studied groups collective graphs of force as a function of "some" displacement (we do not know what the authors finally measured), so that the reader can see the repeatability of the results and draw his own conclusions - the authors should also draw some conclusions.

Table 2 needs to be supplemented with the dimension of the crack - please ensure this.

Paragraph 3.1.1 - please provide the formula for the function Y' - necessarily. What does formula (3b) apply to? – I do not quite understand it – please explain it specifically. In this paragraph, the authors use symbols with a subscript in the formulas, and not in the explanations - whether they are the same symbols or maybe different ones. Some quantities are not explained at all - please check this paragraph for this. Please specify here how the f11(beta) function counts - absolutely. The same remarks apply to paragraph 3.2 - the values and indexes were written incorrectly - formula (4) should be illustrated with an illustrative drawing and a drawing based on the registered signals. Please be sure to complete it.

How the authors estimated the value of KIC, which is considered by many researchers to be a material constant or material property - what were the conditions for recognizing KC as KIC - the same applies to GC and GIC. In the case of steel, there is a condition regarding thickness and other characteristic dimensions - what about rock? Please write it clearly.

In the case of the graphs shown in Figures 2 and 4, please attach graphs relating to the average values determined during experimental studies or the median to the work. For these studies, please attach a statistical analysis of the results obtained - max, min, median, mean, scatter, standard deviation. Please compare the obtained averaged results (or median) with respect to sample loading speed, sample diameter, etc. This is quite important.

The paper lacks information about R2 for a specific fit in some places - especially in the graphs, please complete it.

There are several shortcomings in the present manuscript. Authors should study several similar research papers on a similar topic, see how to present the results, improve the paper, and submit it for re-review.

I recommend a major revision.

Author Response

Dear Reviewer:

We are grateful to your valuable comments and suggestions. According to the comments, we have revised our manuscript. The revised text is in blue color in both the Responses to Review comments and the revised manuscript. In the response, we list all review comments one by one, and our answer to each comment immediately follows each comment.

Yours sincerely,

Yang Qiao, Zong-Xian Zhang and Sheng Zhang

January 6, 2023

Reviewer 2 Report

Here are some major comments to improve the quality of the manuscript. Before addressing these comments, the presented work's quality is insufficient to be published in the journal of Materials.

·               The abstract needs to be rewritten. Do not use a formula in the abstract. The abstract must be a summary of your work including materials, methods, results, etc.

·               It is not needed to mention the number of tested samples in the abstract.

·               In the introduction section, before discussing the formulas and analytical approaches, you need to add a couple of sentences regarding the importance of the topic.

·               There is a lot of research that has been conducted on mode I and II fractures in composite materials which discuss KIC and GIc in different experimental techniques, as well as J-Integral, R-curves, etc. Add a paragraph discussing and briefly presenting the overall methods and techniques to measure fracture toughness. Here is a list of the papers you may add to the introduction section to improve the quality of this section.

Trapezoidal traction–separation laws in mode II fracture in nano-composite and nano-adhesive joints, doi:10.1177/0731684418761001

Analytical investigation on the unstable fracture toughness of fine-grained quartz-diorite rock considering the size effect, https://doi.org/10.1016/j.engfracmech.2022.108722

A cohesive model with a multi-stage softening behavior to predict fracture in nano composite joints, https://doi.org/10.1016/j.engfracmech.2019.106611

·               Clarify the novelty of the work in the last paragraph of the introduction section.

·               Add reference for the data in Table 1.

·               The English writing of the manuscript needs to be reviewed carefully.

·               What is the used standard for the carried out fracture tests? Add the implemented standard to the manuscript.

·               Move the section “3.2. Critical energy-release rate (???)” to the introduction section.

·               Use bullets to present the main achievements of your work in the conclusion section.

·               The paper is more like a technical report than a scientific paper. Rewrite the manuscript.

·               Do not use the words “We”, “Us”, … in a scientific paper. Revise them.

·               The discussion section is more like a literature review than a discussion a scientific work. Rewrite this section.

Author Response

(The authors gave the same response as above.)

Round 2

Reviewer 1 Report

All my suggestions were included by the authors in the revised version of the manuscript. I recommend the paper for publication.

Reviewer 2 Report

The comments and questions are properly answered and modified. The paper is accepted in its current form.